# Evolution of ST-Elevation Acute Myocardial Infarction Prevalence by Gender Assessed Age Pyramid Analysis—The Piramyd Study

**DOI:** 10.3390/jcm7120509

**Published:** 2018-12-03

**Authors:** Aurélie Loyeau, Hakim Benamer, Sophie Bataille, Sarah Tepper, Thévy Boche, Lionel Lamhaut, Virginie Pirès, Benoit Simon, François Dupas, Lisa Weisslinger, Gaëlle Le Bail, Alexandre Allonneau, Jean-Michel Juliard, Yves Lambert, Frédéric Lapostolle

**Affiliations:** 1Registry Department, Regional Health Agency in Great Paris Area, 75008 Paris, France; aurelie.loyeau@sesan.fr (A.L.); s.bataille@orange.fr (S.B.); 2Ramsay Générale de Santé, ICPS, 6 Avenue du Noyer Lambert à Massy, 91300 Massy, France; h.benamer@angio-icps.com; 3Service de cardiologie Hôpital Européen de Paris—La Roseraie, 93300 Aubervilliers, France; 4SAMU 93, UF Recherche-Enseignement-Qualité Avicenne Hospital-APHP, 93000 Bobigny, France; st.sarah.tepper@gmail.com (S.T.); lweisslinger@hotmail.fr (L.W.); 5Université Paris 13, Sorbonne Paris Cité, 75008 Paris, France; 6SAMU 94 Mondor Hospital-APHP, 94000 Créteil, France; thevy.boche@gmail.com (T.B.); lionel@lamhaut.fr (L.L.); 7SAMU de Paris – DAR, Necker Hospital-APHP, 75008 Paris, France; 8Université Paris Descartes, 75008 Paris, France; 9INSERM U 970 équipe 4, 75008 Paris, France; 10SAMU 77, Melun Hospital, 77000 Melun, France; virginie.pires@ch-melun.fr; 11SAMU 91, Sud Francilien Hospital, 91100 Corbeil-Essonnes, France; benoitsimon1981@gmail.com; 12SAMU 95, Pontoise Hospital, 95032 Pontoise, France; francois.dupas@ch-pontoise.fr; 13SAMU 92, Garches Hospital-APHP, 92380 Garches, France; gaelle.le-bail@wanadoo.fr; 14Fire Departement of Paris, EMS Department, 75008 Paris, France; alexandre.allonneau@pompiersparis.fr; 15Cardiology Department, DHU FIRE, Université Paris Diderot, Sorbonne Paris-Cité, INSERM U-1148, Bichat Hospital-APHP, 75008 Paris, France; jean-michel.juliard@aphp.fr; 16SAMU 78, Versailles Hospital, 78150 Le Chesnay, France; yl.samu78@wanadoo.fr

**Keywords:** ST-elevation myocardial infarction (STEMI), prehospital, gender, age, age pyramid

## Abstract

Introduction: Recent studies reported a decrease in the incidence of acute myocardial infarction. This favorable evolution does not extend to young women. The interaction between gender, risk factors and myocardial infarction incidence remains controversial. Objective: To compare the evolution of the age pyramid of patients with ST-elevation myocardial infarction (STEMI) according to gender. Methods: Data from patients with STEMI managed in pre-hospital settings prospectively collected in the greater Paris area. Evolution of patient demographics and risk factors was investigated. Results: 28,249 patients with STEMI were included in the registry between 2002 and 2014, 21,883 (77%) males and 6366 (23%) females. The sex ratio did not significantly vary over the study period (*p* = 0.4). Median patient age was 60.1 years (51.1–73.0) and was significantly different between males and females, respectively 57.9 (50.0–68.3) vs. 72.9 years (58.3–82.2) (*p* = 0.0004). The median age of males significantly (*p* = 0.0044) increased from 57.6 (50.1–70.0) in 2002 to 58.1 years (50.5–67.8) in 2014. The median age of females significantly (*p* = 0.0006) decreased from 73.7 (57.9–81.8) to 69.6 years (57.0–82.4). The median gap between the age of men and women significantly (*p* = 0.0002) decreased, from 16.1 to 11.5 years. Prevalence of risk factors was unchanged or decreased except for hypertension which significantly increased in males. The rate of STEMI without reported risk factors increased in both males and females. Conclusion: The age of STEMI onset significantly decreased in females, whereas it significantly increased in males. The prevalence of risk factors decreased in males, whereas no significant variation was found in females.

## 1. Introduction

Many studies have reported a decrease in the incidence of ST-segment elevation acute myocardial infarction (STEMI) in recent years in Europe and in the United States [1,2]. This finding contrasts with the overall increase in the prevalence of cardiovascular risk factors, particularly diabetes, cholesterol and obesity, and even smoking [3,4,5,6]. However, this trend seems to involve almost all Western countries, all age groups and both genders. Nevertheless, it would appear that this favorable evolution does not extend to young women [7]. The prevalence of risk factors, particularly smoking, is suspected to account for this evolution. This could be particularly true in France where the prevalence of smoking has increased dramatically in women over the last decade, having even doubled in those aged 55–64 [8]. This is important, because smoking is the main determining factor of cardiovascular risk in many young people. Thus, in patients under 55 years of age who had an acute myocardial infarction, smoking was the sole cardiovascular risk factor in 80% of cases [9]. The interaction between risk factors and gender is certainly crucial, but difficult to substantiate. Furthermore, it seemed to us that the analysis of the age pyramid in a registry of acute myocardial infarction that had been running since 2002 would make it possible to study the evolution of the age of onset of acute myocardial infarction.

## 2. Methods

### 2.1. Settings

In France, patients experiencing acute chest pain are urged to rapidly contact the emergency medical services using a nationwide telephone number (15). The medical dispatch centers (SAMUs) which take these calls are uniformly organized in all regional districts and are run by emergency physicians. In cases of suspected acute coronary syndrome, the physician can dispatch a mobile intensive care unit (MICU) manned by an ambulance driver, a nurse trained in emergency medicine, an emergency physician, and in most instances a student. Emergency diagnostic and therapeutic facilities are available on board the MICU, including electrocardiograph (ECG), laboratory test facilities (including troponin assays), ultrasound and drugs (including fibrinolytic agents, antiplatelet agents, anticoagulation, etc.). The decision to perform myocardial reperfusion and the choice of the technique (i.e., prehospital fibrinolysis or primary percutaneous coronary intervention, PCI) are made by the onsite physician in coordination with the medical dispatch center physician and the cardiologist in the unit to which the patient will be admitted. SAMU and MICU are also responsible for the transfer of STEMI patients from hospitals without PCI facilities to cath-labs.

The Greater Paris area includes the city of Paris, three inner suburbs (regional districts) and four outer suburbs. The total area covers a surface of 12,012 km^2^ with a total population of 12 million inhabitants. Each district has its own medical dispatch center (8 SAMUs in total) operating a total of 40 MICUs. Patients are taken to 36 cardiology units with 24/24 PCI facilities.

### 2.2. E-MUST Registry

The e-Must registry contains prospectively collected data on acute myocardial infarction patients handled in prehospital settings by the SAMU of the Greater Paris area. All patients with chest pain lasting less than 24 h, ECG signs of STEMI, and managed by one of the 40 MICU teams are prospectively included in the registry regardless of the severity of the STEMI, and of any administered treatments, including reperfusion strategy (prehospital fibrinolysis or primary PCI) and prehospital and in-hospital outcomes.

Data collected are as follow: patient demographics (age and gender), risk factors, STEMI conditions (infarction location and clinical severity criteria), administered drugs (and time of administration), chosen reperfusion strategy (fibrinolysis or primary PCI) and in-hospital outcome. All the time intervals (chest pain onset/call to the SAMU/first medical contact/treatment/hospital arrival/puncture/balloon) are recorded. Recorded risk factors are: personal coronary history, coronary artery disease in the family, smoking, diabetes, hypertension, dyslipidemia, overweight. Risk factors were identified based on ongoing treatment, drug prescription, previous consultation or hospitalization and self-reported by patient questionnaire. Their definition did not change during the study period. Risk factors have only been included in the registry since 2006.

All patients are informed of their inclusion in the registry which has been registered with the French Data Protection Agency (CNIL—Commission nationale de l’informatique et des libertés) and they receive both oral and written information. The physician in charge of the patient in the prehospital setting checks all data before entry into the registry and carries out a monthly audit. An annual regional audit is conducted under the auspices of the Regional Health Agency (ARS—Agence Régionale de Santé) which posts the results for the region, and for each SAMU and MICU on a dedicated website (http://www.cardio-arsif.org).

### 2.3. Study Design, Data Extraction and Statistical Analysis

This was an observational study based on data extracted from the e-MUST registry for the period 2002–2014. The following data were extracted for analysis: Patient demographics and risk factors.

The evolution of the median age of both male and female populations and the prevalence of risk factors during the period were studied by calculating the correlation coefficient (r) and using the Cochran-Armitage trend test. An *r* value between 0.6 and 0.79 was considered indicative of a strong, downhill or uphill, linear correlation and an r value between 0.8 and 1 indicative of a very strong correlation [10].

The evolution of the age pyramid for male and female populations was evaluated by using the Kolmogorov-Smirnov test. Percentages are used to describe categorical variables and median and inter-quartiles to describes continuous variables. All tests were 2-tailed with the level of statistical significance prespecified at 5% (*p* < 0.05). Statistical analyses were performed using SAS^®^ Software 9.4 (Cary, CA, USA).

## 3. Results

A total of 28,411 patients managed in pre-hospital settings for STEMI were included in the registry between 2002 and 2014. One hundred sixty-two (0.5%) patients were excluded because age (*n* = 93) or gender (*n* = 69) were missing (Figure 1). The median number of patients included was 2168 (2063–2205) per year without significant change during the study period.

Overall, 28,249 patients were included in the analysis, of whom 21,883 (77%) were men and 6366 (23%) women. The median number of patients included was 2168 (2063–2205) per year. The sex ratio did not significantly vary over the period studied (*p* = 0.4 Cochran–Armitage trend test).

The median age of the patients was 60.1 years (51.1–73.0) (*n* = 28,249). It was significantly different between males and females, respectively 57.9 (50.0–68.3) years vs. 72.9 (58.3–82.2) years (*p* = 0.0004) and varied significantly over the course of the study period. The median age of males significantly increased from 57.6 years (50.1–70.0) in 2002 to 58.1 years (50.5–67.8) in 2014 (*p* = 0.0044 Cochran-Armitage trend test) (Figure 2). The median age of females significantly decreased from 73.7 (57.9–81.8) years in 2002 to 69.6 years (57.0–82.4) in 2014 (*p* = 0.0006 Cochran–Armitage trend test) (Figure 2). The evolution is represented on the age pyramid in Figure 2. The median gap between the age of men and women decreased significantly, from 16.1 to 11.5 years (*p* = 0.0002 Cochran-Armitage trend test) (Figure 2).

Significant increase in males (blue circles), from 57.6 years (50.1–70.0) to 58.1 years (50.5–67.8); Cochran-Armitage trend test: *p* = 0.0044. Significant decrease in females (red diamonds), from 73.7 years (57.9–81.8) to 69.6 years (57.0–82.4); Cochran–Armitage trend test: *p* = 0.0006.

Distribution between males and females was significantly different in 2002 and 2014 (*p* < 0.0001, Kolmogorov–Smirnov test). Distribution by quinquennial age was not significantly different between 2002 and 2014 for both females (*p* = 0.07, Kolmogorov–Smirnov test) and males (*p* = 0.15, Kolmogorov–Smirnov test) (Figure 3).

The prevalence of all risk factors differed significantly according to gender (Table 1). In addition, a significant variation in the prevalence of risk factors was observed between 2006 and 2014 with respect to family history of coronary artery disease, excess weight, dyslipidemia and (the) absence of a risk factor (Figure 4). For males, prevalence of excess weight, family history of coronary artery disease, smoking, dyslipidemia decreased significantly and that of hypertension increased significantly. The rate of men without past history (i.e., without any of the systematically recorded risk factors) increased significantly from 6 to 7% (*p* = 0.03). For women, only the prevalence of family history of coronary artery disease decreased significantly (Figure 3). The rate of women without past history of risk factors increased significantly from 9 to 12% (*p* = 0.02). Details of the evolution over the studied period are presented in annexe 1.

The total number of patients who died was 1134 (5.1%). In-hospital mortality rate was significantly higher in females than in males: 485 (8.4%) vs. 851 (4.1%); *p* < 0.0001. In-hospital mortality rate significantly decreased during the study period, in the global population (*p* < 0.0001 Cochran–Armitage trend test) as well as and in males (*p* = 0.008) and females (*p* = 0.02) (Figure 5).

## 4. Discussion

The age of myocardial infarction occurrence significantly decreased in females (74–70 years) whereas it increased significantly, but to a lesser extent in males (from less than 58 years of age to more than 58). The median yearly age difference between males and females decreased dramatically by one third. For males, this evolution can be explained by a decrease in the prevalence of all cardiovascular risk factors (with the exception of arterial hypertension). In contrast, for females, we did not find any significant variation in the prevalence of any single risk factor to explain this evolution. This increase occurred in the context of a one-third increase in patients without risk factors. This observation has a major impact on prevention policies.

An overall trend towards a reduced incidence of acute myocardial infarction has been found in many international studies. Nevertheless, this trend, particularly in France, seemed to be associated with a paradoxical increase in the prevalence of acute myocardial infarction in women under 65 years. The opposite trend was clearly observed in males over the course of this 13-year analysis. A difference of 9 years between median of males and females suffering acute myocardial infarction has been found in an international study. This difference was 7 years in Europe, whereas it was 15 years at the same period in our study [11]. The increase in the prevalence of smoking and, more generally, in the prevalence of risk factors was the first proposed explanation potentially accounting for the trend observed in women. Smoking seemed all the more plausible as it is a major risk factor of acute myocardial infarction onset particularly in young patients. However, this hypothesis was based on weak epidemiological data. The evolution of the prevalence of smoking has been estimated by surveys. Our results clearly show that not only has the prevalence of smoking not increased in women suffering acute myocardial infarction, but that acute myocardial infarction has occurred in an increased number of patients without risk factors. However, the number of patients without any established risk factors remains low. This observation has two major consequences: (1) it suggests another explanation for this evolution and (2) it impacts prevention policies.

The impact of gender on the occurrence, the management and the prognosis of acute coronary syndrome is well known [12]. Overall, females are treated later, benefit less from coronary reperfusion and have higher mortality than males [13,14]. This excess mortality involves multiple confounding factors such as age, prevalence of diabetes or treatment strategies. Multivariate analyses have shown that being a woman remains an independent factor accounting for the lack of reperfusion decisions and excess mortality [15,16]. Such a difference is found even in young women [17].

One of the proposed explanations is the hormonal factor. Even though estrogen protects women against cardiovascular events [18], it may also make them more vulnerable to an acute ischemic event [19]. However, that hormones influence the risk of myocardial infarction in female is debated [20]. In addition, the mechanisms of the acute event could differ, especially in patients without cardiovascular risk factors [21]. Thus, although coronal plaque rupture predominates in males, plaque erosion and microemboli production appear to be more common in females [22]. In contrast, coronary dissection and parietal hematoma are more common in females. This could explain up to 10% of STEMI in females before the age of 50 [23]. Smoking is usually the sole risk factor in young females with acute myocardial infarction [9,24]. Therefore, the relation between smoking, coronary dissection and parietal hematoma should be investigated. Arterial aging could also be different in females [25].

STEMI can occur in patients with few "traditional" cardiovascular risk factors, which is in keeping with our findings. The formation of an intra-mural hematoma from an acquired parietal lesion or not, is, in this case, the basic mechanism of coronary occlusion. The treatment of acute myocardial infarction is currently the same in men and women. In cases of dissection or coronary hematoma, the strategy based on antithrombotic and platelet energetics treatment is no longer suitable. A better knowledge of these physiopathological differences could result in the development of different therapeutic strategies. This could, in turn, influence prevention policies.

Primary (and secondary) prevention of cardiovascular (and neurovascular) events are based on an optimal consideration of risk factors. The marked increase in the occurrence of myocardial infarction without risk factors in increasingly younger women is, in this respect, problematic. A better knowledge of the physiopathological mechanisms of the acute coronary syndrome in women is essential. Consideration of other risk factors can be crucial. Authors recently concluded that the higher risk of myocardial infarction in females could not be explained by established risk factors [20]. Other risk factors such as psychosocial factors, sedentary lifestyle or weather could affect males and females differently [11,26]. In contrast, protective factors such as fruit and vegetable consumption or physical activity are not routinely recorded [11]. Moreover, interactions between risk and protective factors and gender is complex [27]. Special attention should be paid to the preventive education of patients, especially those who are known to carry established risk factors. These patients should quickly enter the care system in order to avoid any further increase in the risk of delayed treatment as well as morbidity and mortality. Dedicated guidelines and education campaigns on the risks specific to women have been developed [28,29]. Physicians should be carefully when managing chest pain in young females with only low or even no risk factors [27]. Reducing sex-based differences in management and outcome of STEMI patients is a modern challenge [30,31].

## 5. Limitations

The first limitation to this work is the lack of recording of risk factors during the first six years of the study period. While this limitation may complicate the analysis, it does not alter the main result of the study. The second limitation concerns the exclusive inclusion of patients who were managed in a prehospital setting by the SAMU-Center 15 system. This represents a rate of approximately 50% of the patients in the region. Variations between these two populations of patients cannot be excluded. Nevertheless, in this analysis, the inclusion of patients secondarily transferred from one institution to another limits the risk of selection bias. How our results can be extrapolated to all the patients with acute coronary syndrome is uncertain. Only patients with ST segment elevation myocardial infarction have been included in the study, while patients with any other feature of acute coronary syndromes are not, currently, included in our registry. However, to the best of our knowledge, STEMI and other acute coronary syndromes are usually considered to be related to identical risk factors. Females have worse outcomes in case of non-STEMI too [32]. Investigations focused on the evolution of risk factors in non-STEMI acute coronary syndrome would be of interest to improve prevention campaigns. Finally, the extrapolation of our results to other regions or even to other countries will need to be confirmed by specific analyses.

## 6. Conclusions

The age of onset of myocardial infarction has decreased by four years in females without any significant variation in the prevalence of risk factors to account for this. In particular, the role of smoking, which is readily incriminated, was not substantiated here. Identifying specific risk factors is crucial to optimizing prevention policies.

## Figures and Tables

**Figure 1 jcm-07-00509-f001:**
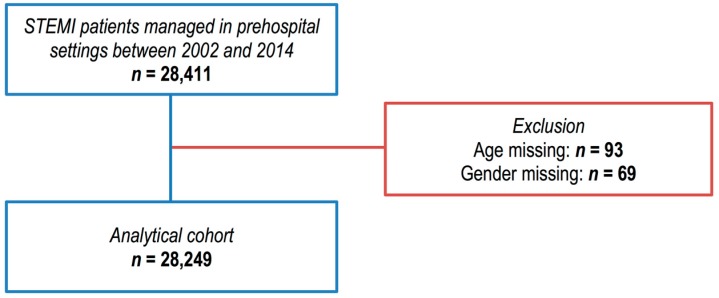
Strengthening the reporting of observational studies in epidemiology (STROBE) diagram of exclusions of cases from the Greater Paris area E-MUST registry.

**Figure 2 jcm-07-00509-f002:**
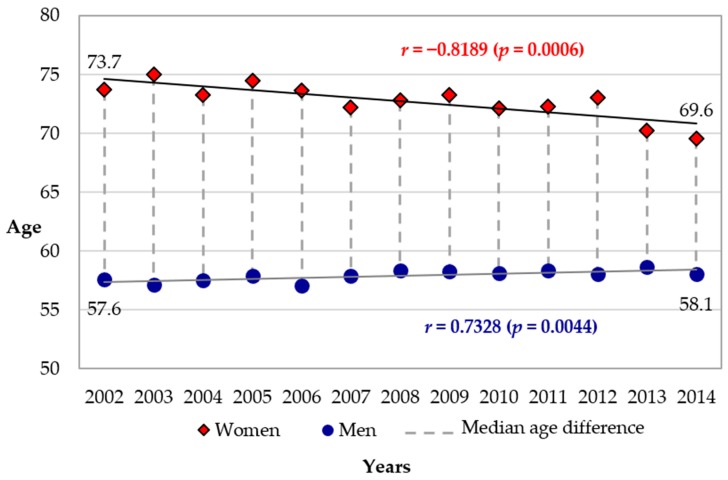
Evolution of the median age of STEMI patients by gender from 2002 to 2014 (*n* = 28,249). *r*, regression correlation coefficient.

**Figure 3 jcm-07-00509-f003:**
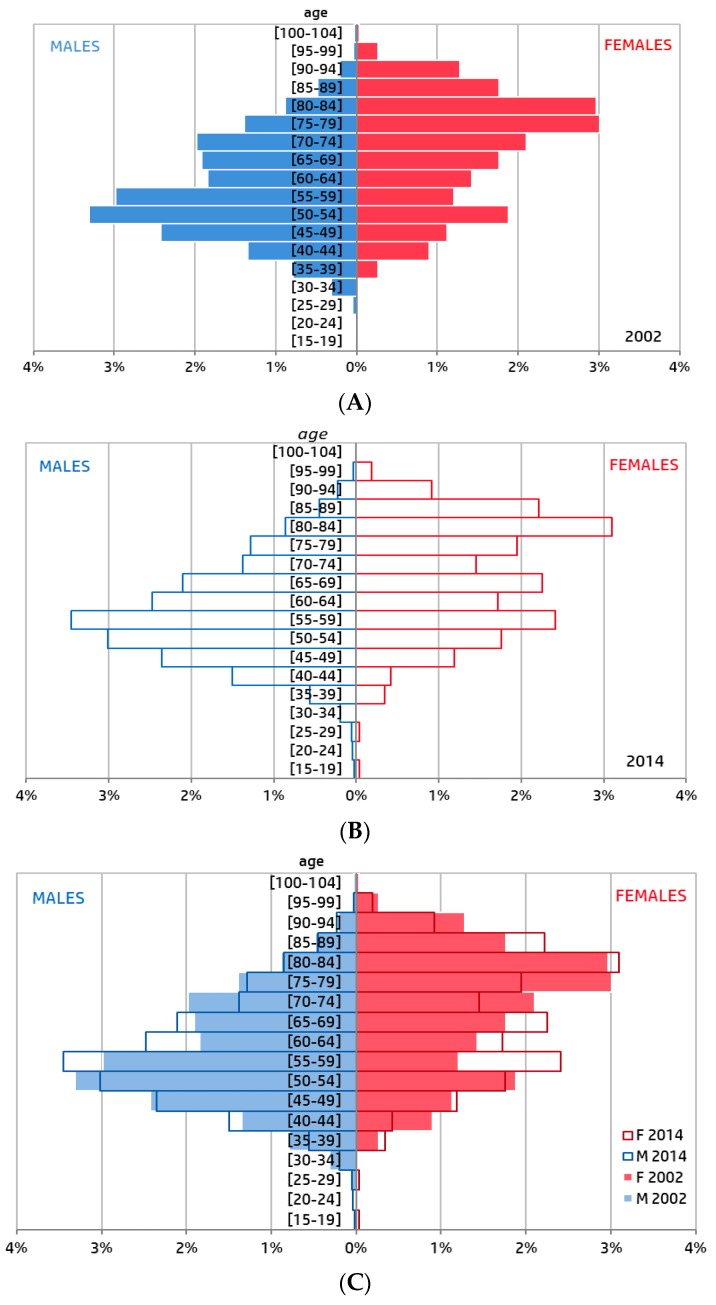
Comparison of the age pyramid of ST-elevation myocardial infarction (STEMI) patients by gender. (**A**) Years 2002 (*n* = 2205); (**B**) Year 2014 (*n* = 2349); (**C**) Years 2002 and Year 2014.

**Figure 4 jcm-07-00509-f004:**
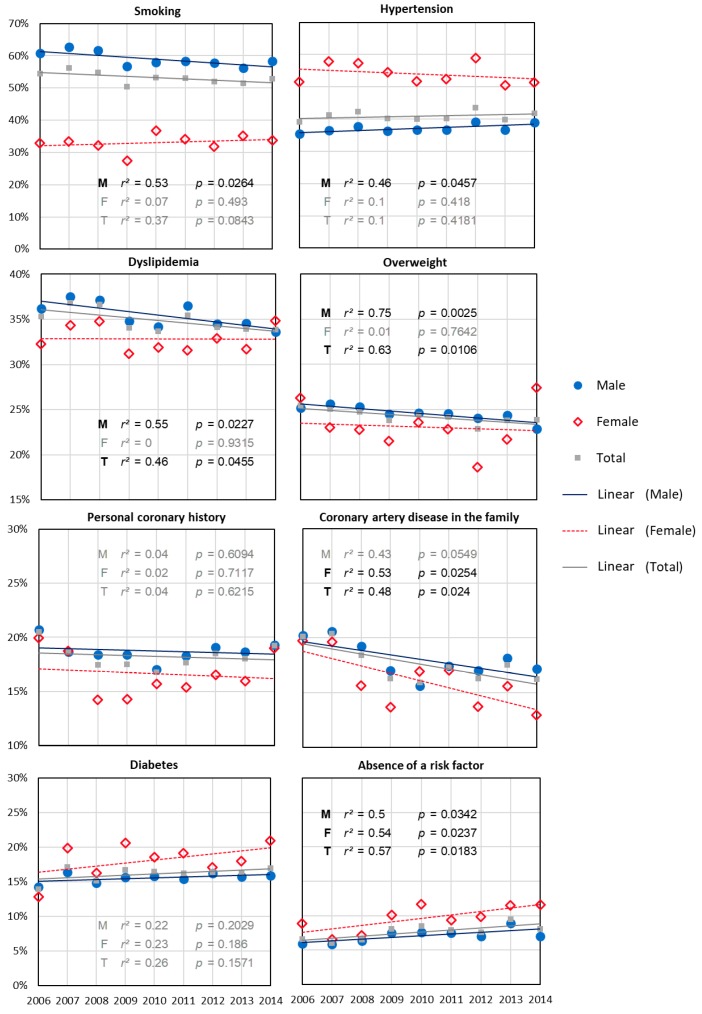
Evolution of the prevalence of risk factors, in the general population, and in males and females, from 2006 to 2014 (*n* = 19,684).

**Figure 5 jcm-07-00509-f005:**
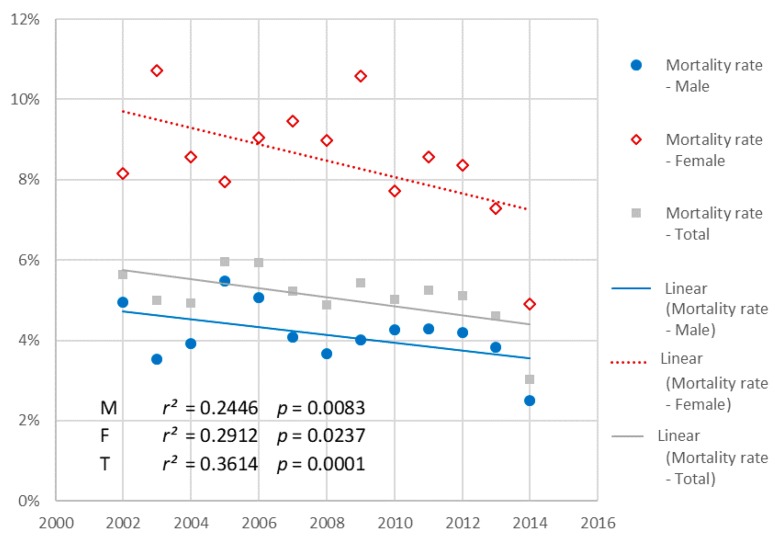
Evolution of the prevalence of risk factors, in the general population, and in males and females, from 2006 to 2014 (*n* = 20,758).

**Table 1 jcm-07-00509-t001:** Comparison of the prevalence of risk factors according to gender (*n* = 19,684).

	*n*	Men	Women	*p*-Value (khi2 Test)
Personal coronary history	19,077	2772 (19%)	709 (17%)	0.0015
Family coronary artery disease	19,076	2664 (18%)	681 (16%)	0.0019
Smoking	19,018	8686 (59%)	1406 (33%)	<0.0001
Diabetes	19,076	2309 (16%)	777 (18%)	<0.0001
Hypertension	19,078	5507 (37%)	2306 (54%)	<0.0001
Dyslipidemia	19,079	5242 (35%)	1404 (33%)	0.0021
Excess weight	19,082	3633 (25%)	982 (23%)	0.0388
Absence of risk factor	19,076	1054 (7%)	413 (10%)	<0.0001

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
