# Peer review of "Evolution of ST-Elevation Acute Myocardial Infarction Prevalence by Gender Assessed Age Pyramid Analysis—The Piramyd Study"

_jcm, 2018, doi:10.3390/jcm7120509_

Reviewer 1 Report

General comments:

 This is an interesting work on an important topic in the cardiovascular arena. Acute myocardial infarction and especially STEMI are strongly associated with premature death and increased mortality and morbidity. This work studied the period of 12 years (from 2002 until 2014) and included more than 28,000 patients from greater Paris area that had STEMI, while the subset of patients that had data on risk factors included timeframe from 2006 until 2014 (8 years). This is a substantial sample to make epidemiological conclusions for this specific region of the world. Results suggest that the median age of STEMI event increased by nearly one year in males while it decreased by more than 4 years. Furthermore, the median age gap between women and men decreased by nearly 5 years.

The manuscript is generally well-written, however, there are areas that could be further improved and are elaborated in the specific comments below.

Legal and ethical disclosures are appropriately addressed in the manuscript.

Statistical methods in this manuscript are appropriate.

References are appropriate, however, they should be inserted within the sentence, not after the dot mark. Please amend that throughout the manuscript.

Specific comments:

1.       Since this analysis dealt with AMI patients that had STEMI, this should be clarified in the title of the manuscript, therefore, it should state „acute ST-segment elevation myocardial infarction“ instead of „acute myocardial infarction“. Please change this throughout the manuscript. It is important to keep clear nomenclature since AMI conventionally also includes NSTEMI while ACS includes STEMI + NSTEMI + unstable AP.

2.       Figure 3. Please divide this image in parts A, B, C and label and explain these properly in the Figure legend.

3.       Figure 4. Please define and explain labels H, F, and T in the Figure legend.

4.       I think that this manuscript could be largely benefiting by including the additional figure (similar to Figure 2) that could show the incidence of STEMI per year (2002 and onward). This is important since the reader could then get a quick grasp on the dynamics and eventual trends of STEMI incidence in your region over the period of 12 years.

5.       What was the in-hospital mortality in STEMI by analyzed year? It would be beneficial to know and see the trend of in-hospital mortality across this 12 year period.

6.       One of the key questions is what are the clinical outcomes between sexes since this is sex-specific and gender-driven analysis. Was there a significant difference regarding in-hospital mortality between sexes? It would be worthwhile to include this in the analysis and to show how this mortality changed over the years (if it did) and was there a gap between women and men? While stratification of sexes by age is a relevant endeavor, it is crucial to see how these are translated in hard clinical endpoints such as in-hospital mortality. Since this was included in your registry, as stated in „Methods“ section I would advise authors to shed more information and light on this issue.

7.       Line 191. Please reformat this sentence and remove the word „this“ before STEMI.

8.       In the „Methods“ section please define which cut-off was used to determine arterial hypertension, how did you define dyslipidemia (through a questionnaire or through lab results) and also clarify how did you define patients that were overweight and had a history of coronary artery disease. Please cite relevant guidelines that substantiate these cut-offs and criteria that were used in your registry.

9.       It is a well-known fact that women, on average, present significantly later with an ACS compared to men and this difference is 7 to 10 years in most of the studies. In your study, you revealed that this gap between sexes is becoming more narrow and it could not be explained by the changes in traditional risk factors (if there was a change at all). I think that this is a particularly important highlight of this study and the reasons why median age in women decreased for 4 years over time should be discussed to a greater detail and substantiated with relevant references from this field. In this light, authors should try to compare their results with relevant trends and dynamics in other countries from Europe, US, Asia and similar findings worthwhile.

10.   STEMI is an ECG diagnosis, with clear reperfusion guidelines and straight drive to a cath lab or fibrinolysis if primary PCI is not feasible within 120 minutes from the time of diagnosis. Therefore, it is to expected that eventual differences in treatment would be lesser between men and women in the STEMI setting and this would be much more prominent in cases of NSTEMI or unstable angina. Authors did a good job in emphasizing different mechanisms of the ACS in women vs. men and this is very important. Besides traditional risk factors that did not account for this change, what are other factors that might drive earlier presentation of ACS and STEMI in women?

11.   I would advise authors to substantiate their work with more relevant references from this field and to cite proper guidelines that were employed. Nineteen references are not acceptable for the original manuscript.

 Author Response

Reviewer 1

This is an interesting work on an important topic in the cardiovascular arena. Acute myocardial infarction and especially STEMI are strongly associated with premature death and increased mortality and morbidity. This work studied the period of 12 years (from 2002 until 2014) and included more than 28,000 patients from greater Paris area that had STEMI, while the subset of patients that had data on risk factors included timeframe from 2006 until 2014 (8 years). This is a substantial sample to make epidemiological conclusions for this specific region of the world. Results suggest that the median age of STEMI event increased by nearly one year in males while it decreased by more than 4 years. Furthermore, the median age gap between women and men decreased by nearly 5 years.

The manuscript is generally well-written, however, there are areas that could be further improved and are elaborated in the specific comments below.

Legal and ethical disclosures are appropriately addressed in the manuscript.

Thank you very much for these kind comments.

Statistical methods in this manuscri 1.       Since this analysis dealt with AMI patients that had STEMI, this should be clarified in the title of the manuscript, therefore, it should state „acute ST-segment elevation myocardial infarction“ instead of „acute myocardial infarction“. Please change this throughout the manuscript. It is important to keep clear nomenclature since AMI conventionally also includes NSTEMI while ACS includes STEMI + NSTEMI + unstable AP.

This correction has been made.  

2.       Figure 3. Please divide this image in parts A, B, C and label and explain these properly in the Figure legend.

This correction has been made. As suggested by the reviewer 2, colors were added in the figures to facilitate analysis.    

3.       Figure 4. Please define and explain labels H, F, and T in the Figure legend.

The correction has been made. We forgot to translate this part of the legend… H was for Male: Homme in French… As suggested by the reviewer 2, colors were added in the figure to facilitate analysis.    

4.       I think that this manuscript could be largely benefiting by including the additional figure (similar to Figure 2) that could show the incidence of STEMI per year (2002 and onward). This is important since the reader could then get a quick grasp on the dynamics and eventual trends of STEMI incidence in your region over the period of 12 years.

The result has been added in the beginning of the results section. As no significant change has been reported during the study period, we didn’t add any figure.

5.       What was the in-hospital mortality in STEMI by analyzed year? It would be beneficial to know and see the trend of in-hospital mortality across this 12 year period.

Mortality rate during the study period significantly decreased in males and females. This result has been added in the results section (and presented in the methods section). A figure has been added to this evolution.  

6.       One of the key questions is what are the clinical outcomes between sexes since this is sex-specific and gender-driven analysis. Was there a significant difference regarding in-hospital mortality between sexes? It would be worthwhile to include this in the analysis and to show how this mortality changed over the years (if it did) and was there a gap between women and men? While stratification of sexes by age is a relevant endeavor, it is crucial to see how these are translated in hard clinical endpoints such as in-hospital mortality. Since this was included in your registry, as stated in „Methods“ section I would advise authors to shed more information and light on this issue.

This result has been added in the results section. In hospital mortality rate was significantly different in males and females.   

7.       Line 191. Please reformat this sentence and remove the word „this“ before STEMI.

This correction has been made.

8.       In the „Methods“ section please define which cut-off was used to determine arterial hypertension, how did you define dyslipidemia (through a questionnaire or through lab results) and also clarify how did you define patients that were overweight and had a history of coronary artery disease. Please cite relevant guidelines that substantiate these cut-offs and criteria that were used in your registry.

These points have been clarified in the methods section.

9.       It is a well-known fact that women, on average, present significantly later with an ACS compared to men and this difference is 7 to 10 years in most of the studies. In your study, you revealed that this gap between sexes is becoming more narrow and it could not be explained by the changes in traditional risk factors (if there was a change at all). I think that this is a particularly important highlight of this study and the reasons why median age in women decreased for 4 years over time should be discussed to a greater detail and substantiated with relevant references from this field. In this light, authors should try to compare their results with relevant trends and dynamics in other countries from Europe, US, Asia and similar findings worthwhile.

This point has been underlined in the (enlarged) discussion section. Bibliography has been up-dated.  

10.   STEMI is an ECG diagnosis, with clear reperfusion guidelines and straight drive to a cath lab or fibrinolysis if primary PCI is not feasible within 120 minutes from the time of diagnosis. Therefore, it is to expected that eventual differences in treatment would be lesser between men and women in the STEMI setting and this would be much more prominent in cases of NSTEMI or unstable angina. Authors did a good job in emphasizing different mechanisms of the ACS in women vs. men and this is very important. Besides traditional risk factors that did not account for this change, what are other factors that might drive earlier presentation of ACS and STEMI in women?

This is discussed in the (enlarged) discussion section. Bibliography has been up-dated.  

11.   I would advise authors to substantiate their work with more relevant references from this field and to cite proper guidelines that were employed. Nineteen references are not acceptable for the original manuscript.

Recent references have been added and the discussion section developed. Bibliography has been up-dated.      

References are appropriate, however, they should be inserted within the sentence, not after the dot mark. Please amend that throughout the manuscript.

This correction has been made.

Reviewer 2 Report

1)      It is an observational study that has included a large number of patients over a period of 13 years.

The topic is remarkable and the results are in part in line with epidemiological data on the incidence of acute myocardial infarction in women, in part differing from what is known in relation to the more conventional risk factors for coronary heart disease and therefore raise some questions.

However, evolving sex-specific research has demonstrated that although men and women share similar risk factors for CHD, certain risk factors are more potent in women. These also include non-conventional factors. Among these, the Authors should discuss the possible role of psychosocial risk factors, including depression, psychological and emotional stress.

2)      The authors should improve Figure 4, which overall is not clear; the possible use of different colors in addition to the different dashes could help.

3)      Authors should check the spelling of the text and eliminate some repetitions.

Author Response

 Reviewer 2

1)      It is an observational study that has included a large number of patients over a period of 13 years.

The topic is remarkable and the results are in part in line with epidemiological data on the incidence of acute myocardial infarction in women, in part differing from what is known in relation to the more conventional risk factors for coronary heart disease and therefore raise some questions.

Thank you very much for these kind comments.

However, evolving sex-specific research has demonstrated that although men and women share similar risk factors for CHD, certain risk factors are more potent in women. These also include non-conventional factors. Among these, the Authors should discuss the possible role of psychosocial risk factors, including depression, psychological and emotional stress.

As suggested, this point has been added in the (enlarged) discussion section. Bibliography has been up-dated.  

2)      The authors should improve Figure 4, which overall is not clear; the possible use of different colors in addition to the different dashes could help.

The figure has been modified, colors have been introduced. We trust this revised figure is clear.  

3)      Authors should check the spelling of the text and eliminate some repetitions.

The text has been carefully reviewed to correct the spelling and limit repetition.

Reviewer 3 Report

This is an interesting study, that investigates the evolution of the age pyramid of patients with STEMI according to gender based on prospectively collected pre-hospital data (e-Must registry) from the Greater-Paris area for the period 2002 to 2014. Several findings are very interesting, particularly that there is an increase of proportion of patients with no previous history of coronary disease and no risk factors. This might indicate that the risk factors for coronary artery disease go beyond the known factors, and need profound investigation in basic, clinical and population studies.

·        The authors stated, that ‘This favorable evolution does not extend to young women.’ How did they define ‘young women’? What did they find regarding young women? This sentence seems detached from the remaining abstract, since the proportion of young (according to the text<65 years) cases is not mentioned in the abstract.

·        Please clarify, does ‘without past history’ mean ‘without personal history of coronary disease’? (lines 144, and 145-146). Could the authors also comment on the increase of the proportion of men and women without past history of coronary disease?

·        Line 165. Please correct the ‘15-year analysis’ to ‘13 year analysis.’

·        The discussion is missing several references. One example is this sentence; ‘Smoking is usually the sole risk factor in 188 young females with acute myocardial infarction. ‘ Line 188-189. The part on MI pathophysiology is interesting though.

·        One limitation of the study is that only STEMI patients have been included. It would be interesting to know whether there have been changes in NSTEMI populations. The NSTEMI population could be potentially changed due to possible changes in the time to first medical contact after chest pain onset related to improved response times. Could the authors acknowledge this as a limitations?

Author Response

Reviewer 3

This is an interesting study, that investigates the evolution of the age pyramid of patients with STEMI according to gender based on prospectively collected pre-hospital data (e-Must registry) from the Greater-Paris area for the period 2002 to 2014. Several findings are very interesting, particularly that there is an increase of proportion of patients with no previous history of coronary disease and no risk factors. This might indicate that the risk factors for coronary artery disease go beyond the known factors, and need profound investigation in basic, clinical and population studies.

Thank you very much for these kind comments. We agree with the reviewer (and this a major result of our study) that ‘’unknown’’ (currently non-identified risk factors) should be investigated.

·        The authors stated, that ‘This favorable evolution does not extend to young women.’ How did they define ‘young women’? What did they find regarding young women? This sentence seems detached from the remaining abstract, since the proportion of young (according to the text<65 years) cases is not mentioned in the abstract.

This sentence has been modified. This study didn’t focus on ‘’young women’’, so such a conclusion was inappropriate.  

·        Please clarify, does ‘without past history’ mean ‘without personal history of coronary disease’? (lines 144, and 145-146). Could the authors also comment on the increase of the proportion of men and women without past history of coronary disease?

It means without known risk factors (including personal history of coronary disease). This has been clarified in the methods section.  

·        Line 165. Please correct the ‘15-year analysis’ to ‘13 year analysis.’

The correction has been made.

·        The discussion is missing several references. One example is this sentence; ‘Smoking is usually the sole risk factor in 188 young females with acute myocardial infarction. ‘ Line 188-189. The part on MI pathophysiology is interesting though.

The reference has been added. As suggested, the discussion section has been enlarged. Bibliography has been up-dated.  

·        One limitation of the study is that only STEMI patients have been included. It would be interesting to know whether there have been changes in NSTEMI populations. The NSTEMI population could be potentially changed due to possible changes in the time to first medical contact after chest pain onset related to improved response times. Could the authors acknowledge this as a limitations?

We agree with the reviewer. Our registry doesn’t include NSTEMI patients. This (potential) limitation has been added in the dedicated section.

Round  2

Reviewer 1 Report

Authors have addressed my concerns effectively.